# Thousands of *Pristionchus pacificus* orphan genes were integrated into developmental networks that respond to diverse environmental microbiota

Marina Athanasouli, Nermin Akduman[¤], Waltraud Röseler, Penghieng Theam, Christian Rödelsperger[ID]*

Department for Integrative Evolutionary Biology, Max Planck Institute for Biology, Tübingen, Germany

¤ Current address: Interfaculty Institute of Microbiology and Infection Medicine, University of Tübingen, Tübingen, Germany
* christian.roedelsperger@tuebingen.mpg.de

**Data Availability Statement:** Raw reads were deposited at the European Nucleotide archive under the study accession PRJEB60166.

## Abstract

Adaptation of organisms to environmental change may be facilitated by the creation of new genes. New genes without homologs in other lineages are known as taxonomically-restricted orphan genes and may result from divergence or *de novo* formation. Previously, we have extensively characterized the evolution and origin of such orphan genes in the nematode model organism *Pristionchus pacificus*. Here, we employ large-scale transcriptomics to establish potential functional associations and to measure the degree of transcriptional plasticity among orphan genes. Specifically, we analyzed 24 RNA-seq samples from adult *P. pacificus* worms raised on 24 different monoxenic bacterial cultures. Based on coexpression analysis, we identified 28 large modules that harbor 3,727 diplogastrid-specific orphan genes and that respond dynamically to different bacteria. These coexpression modules have distinct regulatory architecture and also exhibit differential expression patterns across development suggesting a link between bacterial response networks and development. Phylostratigraphy revealed a considerably high number of family- and even species-specific orphan genes in certain coexpression modules. This suggests that new genes are not attached randomly to existing cellular networks and that integration can happen very fast. Integrative analysis of protein domains, gene expression and ortholog data facilitated the assignments of biological labels for 22 coexpression modules with one of the largest, fast-evolving module being associated with spermatogenesis. In summary, this work presents the first functional annotation for thousands of *P. pacificus* orphan genes and reveals insights into their integration into environmentally responsive gene networks.

## Author summary

The inference of biological function for genes in newly sequenced genomes heavily relies on sequence conservation with classical model organisms. Consequently, newly evolved

**Funding:** This work was funded by the Max Planck Society. The funder had no role in study design, data collection and analysis, decision to publish, or preparation of the manuscript.

**Competing interests:** The authors have declared that no competing interests exist.

orphan genes that do not have homologs will not be associated with any function. Here, we use coexpression with known genes in order to assign potential functions to orphan genes in the nematode *Pristionchus pacificus*. To this end, we generated transcriptome profiles of *P. pacificus* worms that were grown on 24 different bacteria and clustered the genes into 28 large coexpression modules which contain thousands of orphan genes. Integrative analysis could associate most coexpression modules with biological processes or tissues, which results in the first functional annotation for thousands of orphan genes. Complementary analysis of gene ages shows that modules associated with female reproduction are highly constrained whereas a male-reproductive module is much more likely to integrate new genes, which could possibly be explained by sperm competition. This links sexual conflict with gene network evolution and environmental regulation.

## Introduction

The evolution of phenotypic diversity across all domains of life has been accompanied by the formation of new genes. As a result, up to one third of the gene content in extant genomes consists of orphan genes that lack homologs in other taxonomic lineages [1]. Although the definition of these taxonomically-restricted orphan genes is context-dependent and the numbers of identified orphan genes will vary with the phylogenetic resolution and methods for homology detection, orphan genes exist in any given genome [2]. Orphan genes may reflect ancient genes that evolve so fast that homology cannot be detected [3]. Alternatively, orphan genes may arise *de novo* from previously non-coding sequences [4,5]. This has been demonstrated in multiple taxonomic groups including vertebrates [6,7], insects [8,9], nematodes [10,11], yeast [12,13], and plants [14,15]. Horizontal gene transfer can be considered as a third mechanism that would lead to the generation of orphan genes [16,17]. The origin of orphan genes can most conclusively be studied for relatively recent gene births because potentially neutrally evolving ancestral sequences may degenerate very fast in sister taxa rendering homology detection impossible [18]. Thus, to what extent more ancient orphan genes arose from divergence or *de novo* formation is currently not known. Another important question concerns the biological functions of orphan genes. Given that large-scale experimental screens for possible functions are only possible in a limited number of model species [19,20] and that inference of function based on sequence conservation is not possible, most orphan genes have no known function. It has been hypothesized that new genes play a role in adapting to changing environments [21], but this is only supported by limited data [22,23]. In this study, we want to test this hypothesis in the nematode model organism *Pristionchus pacificus* by exploring the transcriptomic changes after exposure to different bacteria. *P. pacificus* is a free-living nematode that was initially established as a model system for comparative studies with the classical model organism *Caenorhabditis elegans* [24]. *C. elegans* and *P. pacificus* were estimated to have shared the last common ancestor around 130–310 million years ago [25]. Numerous studies have revealed conserved and divergent patterns across development [26], neurobiology [27], and behavior [28]. When the genome of *P. pacificus* was sequenced, around one third of its genes were classified as orphan genes that do not have any homolog outside the diplogastrid family [29]. By sequencing nine additional diplogastrid genomes, we established a phylogenomic framework to assign these orphan genes into phylostrata and to study their evolutionary dynamics and origin at the inter- and intra-species level [30,31]. These studies confirmed trends such as little expression evidence of orphan genes, rapid evolution, and high turnover which were also found in genomic studies of other animals and plants [32,33]. In contrast to

the broad understanding of new gene evolution in *Pristionchus* nematodes, only two orphan genes have been experimentally characterized. The first orphan gene, *dauerless*, was identified in a screen to dissect the genetic basis of natural variation in dauer formation [34]. The dauer stage represents an alternative developmental stage in most nematodes that allows the worms to survive unfavorable environmental conditions for extended time periods. Overexpression of *dauerless* results in the suppression of dauer formation and it has been speculated that the ability to regulate *dauerless* dosage might provide individual strains an advantage during intraspecific competition [34]. The second orphan gene, *self-1*, encodes a micropeptide that was identified in a screen for killing behavior. Knockouts or modifications of *self-1* resulted in a loss of self-recognition and led to killing by relatives. These two cases demonstrate that *Pristionchus* orphan genes are involved in important developmental decisions and in the evolution of novel behaviors. To complement this detailed functional knowledge of a very limited number of orphan genes with broader functional data on a genome-wide scale, we employ expression data as a proxy for function. This is under the assumption that in order to carry out a certain function, a gene must be expressed at a given condition. Furthermore, large-scale expression data can be used to group genes into functional modules and to transfer functional annotations based on coexpression [35–37]. Specifically, we aim to investigate the *P. pacificus* transcriptome in response to different microbiota which denote the assemblage of microorganisms present in a defined environment [38]. Environmental bacteria can interact with nematodes in a variety of different ways such as serving as food source [39], constituents of the gut microbiome [40], or pathogens [41]. Here, we focus on the transcriptomic response to 24 different bacteria, most of which were isolated previously from *Pristionchus*-associated environments [42]. These bacteria are non-pathogenic in the sense that worm populations can survive for at least several days [42]. This is in contrast to highly pathogenic strains that can kill complete worm populations within a few hours [41]. Our main goals are to group *P. pacificus* genes into functional modules that respond differently to environmental microbiota, to characterize these modules, and finally to test whether some of these modules are enriched in orphan genes. This will allow us to better understand the plastic response to diverse environmental microbiota and to further elucidate the regulation and evolution of the associated gene networks.

## Results

### The transcriptomic response of nematodes to various microbiota does not strictly reflect bacterial phylogeny

To investigate the transcriptomic response of *P. pacificus* to different environmental microbiota, we grew worms on monoxenic cultures of 24 bacterial strains that included commonly used food bacteria such as *Escherichicha coli* OP50 and HB101 as well as 22 bacterial strains that were previously isolated from *Pristionchus*-associated environments [42]. The selected bacteria represent Alpha-, Beta-, Gammaproteobacteria as well as Flavobacteriia. A single RNA-seq data was generated per bacterial strain from 50 young adult worms that were manually picked from mixed-stage cultures (see *Methods*). Most transcriptome profiles of worms grown on different bacteria appeared to be highly similar (Pearson r > 0.9, Fig 1A). One sample, *Wautersiella* LRB104, showed a quite distinct transcriptome with correlation coefficients around 0.8 (Fig 1A and 1B). However, even this outlier shows much higher correlations than transcriptomes from different developmental stages with correlation coefficients of 0.6 [43,44]. The observed correlation coefficients translate into hundreds to thousands of genes with an absolute fold change > 2 between the most similar and most dissimilar pair of samples (S1 Fig). Next, we wanted to test whether the transcriptome profiles follow a phylogenetic pattern,

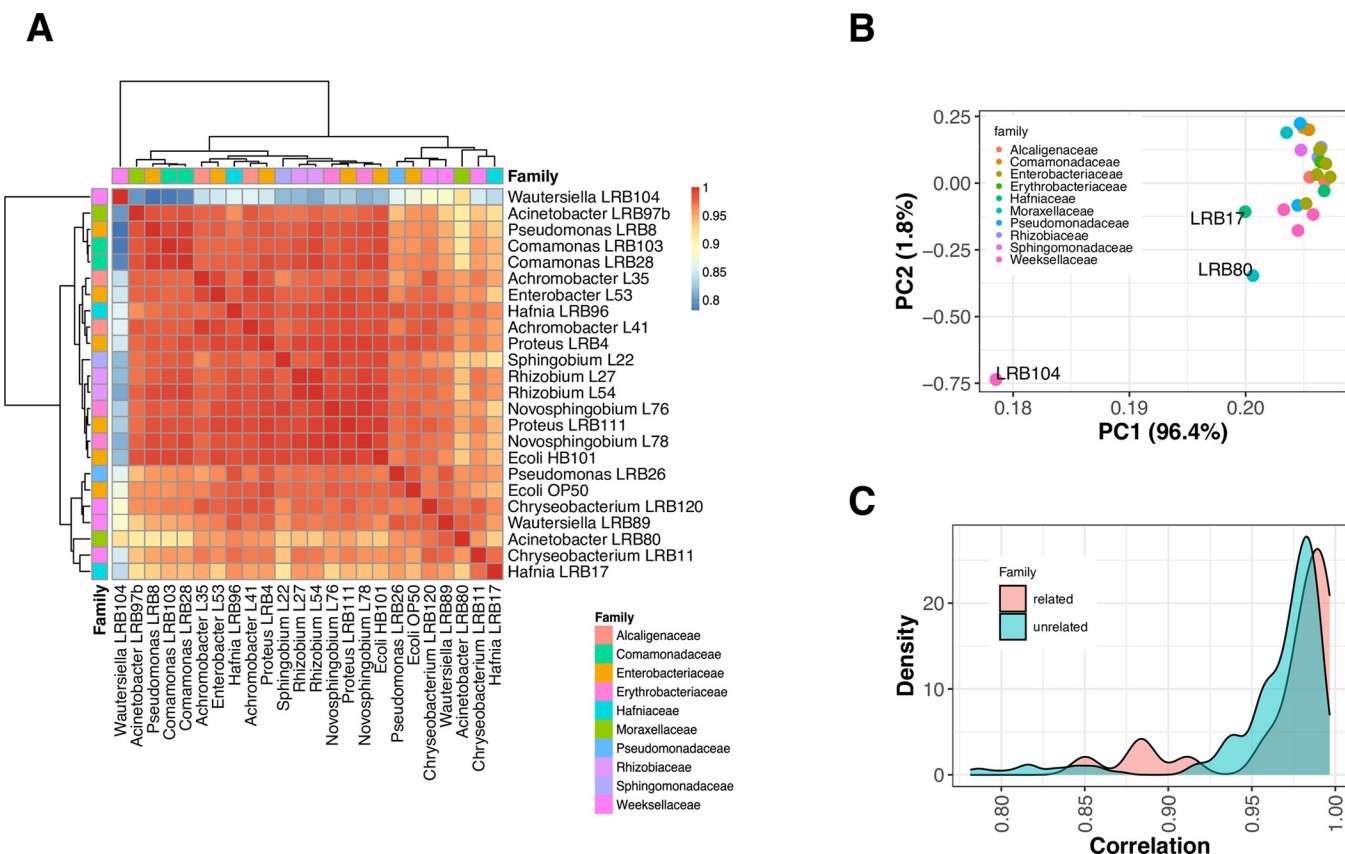

**Fig 1. Transcriptional response to 24 bacterial environments.** (A) The heatmap shows the correlation between transcriptomes. Apart from *Wautersiella* LRB104 all transcriptomes are highly similar. (B) Complementary analysis using principal component analysis identifies *Wautersiella* LRB104 as the most distinct environment. C) The histograms show the distribution of correlation values for comparisons within and across bacterial families. The expression profiles of *P. pacificus* worms on bacteria of different families can be more similar than the profile on bacteria of the same family. This suggests that the transcriptional response does not strictly reflect bacterial phylogeny.

i.e. are transcriptome profiles of worms grown on bacteria of the same family more similar than nematode transcriptomes from more distantly related bacteria (Fig 1C). This did not show significantly higher correlation for transcriptomes from more closely related bacteria (P = 0.35, t-test) suggesting that the transcriptomic response of nematodes to various environmental microbiota do not strictly reflect phylogeny. This observation could indicate that the observed differences are driven by factors that are difficult to control (e.g. bacterial concentration), or alternatively, that strain-specific changes are obscuring family-specific signals. The latter would be the case if critical metabolic pathways are plasmid encoded and could easily be horizontally transferred. In summary, *P. pacificus* nematodes exhibit substantial transcriptomic variation in response to environmental microbiota and these responses do not strictly reflect the bacterial phylogeny.

## Almost half of all genes respond to diverse environmental microbiota

Nematodes interact with bacteria in many different ways involving processes like chemical attraction and repulsion [45], digestion [39], and detoxification [46]. Moreover, those processes can trigger secondary effects on worm development and physiology [47]. Thus, the response of worms might involve multiple regulatory and metabolic pathways and might impact many other biological processes. To characterize gene networks that respond to diverse

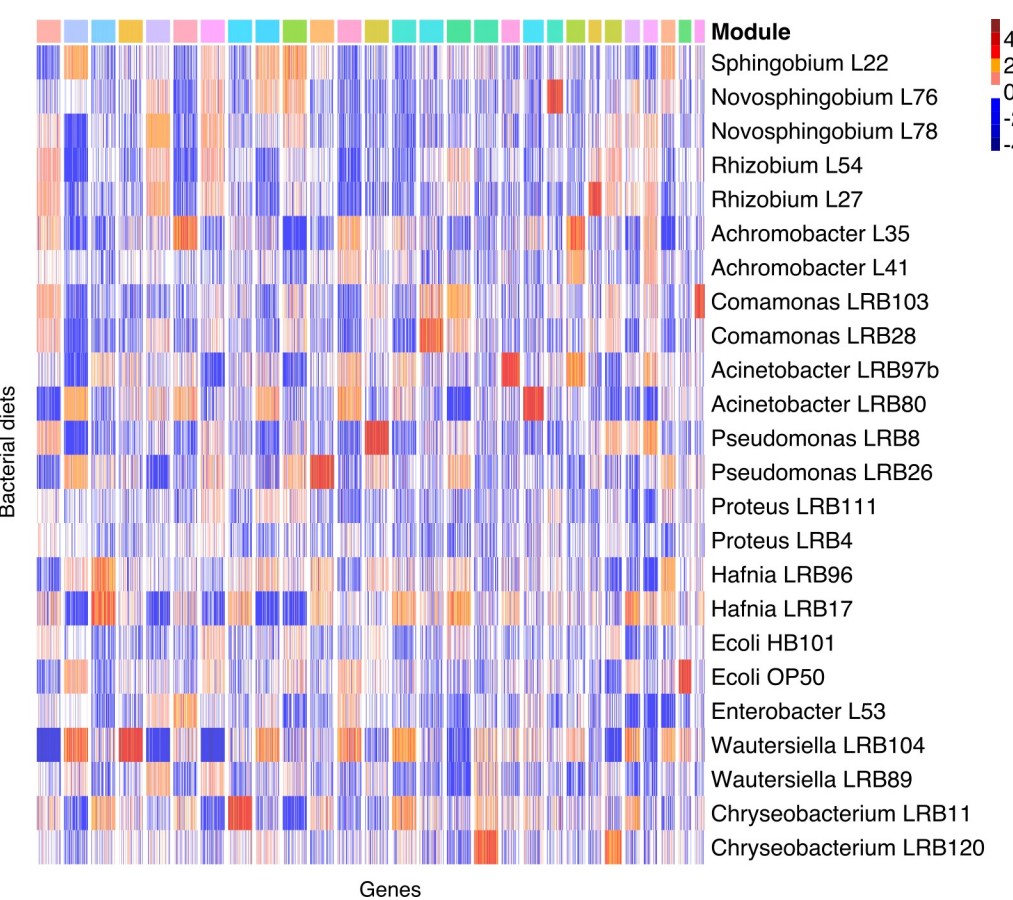

**Fig 2. Expression level of environmentally responsive coexpression modules.** We visualized the z-score normalized expression levels for the 28 largest modules across the bacterial environments. Modules with more than 100 genes were randomly downsampled to 100 genes. While genes of module 4 are most strongly expressed on *Wautersiella* LRB104, module 3 shows the highest expression in response to the two *Hafnia* strains. This demonstrates that environmental microbiota can modulate specific coexpression modules.

environmental microbiota, we computed coexpression modules from the RNA-seq data using a widely used graph clustering approach [48]. Based on correlation coefficient of 0.7 (S2 Fig), we identified 28 large coexpression modules with more than 50 genes (Fig 2 and S1 Table). The largest modules exhibit relatively drastic changes on only one or few bacteria (mostly *Wautersiella* LRB104 and *Hafnia* LRB17), but also more subtle differences on other bacteria. For example, genes of module 1 show strongly reduced expression on *Wautersiella* LRB104, but also mildly lower expression on some other bacteria. These bacteria yield largely opposite trends for genes in module 2. Module 3 shows highest expression on the *Hafnia* strains and the strongest pattern of module 4 results again from high expression on *Wautersiella* LRB104. Also, module 5 shows weak expression on *Wautersiella* LRB104, *Pseudomonas* LRB26, and *Hafnia* LRB17, whereas module 6 stands out by having high expression on *Achromobacter* L35. Given that the most extreme expression differences are frequently observed in the transcriptomic response to *Wautersiella* LRB104, we compared developmental timing between *E. coli* OP50 and both *Wautersiella* strains (LRB89 and LRB104). This demonstrated that worms exhibit a developmental delay on both *Wautersiella* bacteria relative to *E. coli* OP50 (S3 Fig). Given that both *Wautersiella* strains change the developmental timing of *P. pacificus* worms, we cannot fully explain why the transcriptomic response to *Wautersiella* LRB104 differs so

starkly from all the other transcriptomes. To test, how strongly this outlier influences the structure of the coexpression network and the subsequent analysis, we recomputed coexpression modules after removing the *Wautersiella* LRB104 data and compared coexpression modules across both data sets (S4 Fig). This showed a clear one-to-one correspondence between most coexpression modules indicating that the structure of the coexpression network is largely robust. In addition, systematic analysis of coexpression networks that were computed from subsampled data revealed that with fewer RNA-seq samples, network modules tend to get larger (S5 Fig). This may be due to two reasons. First, with fewer samples, it is easier to exceed a given correlation threshold just by chance. Second, with higher numbers of RNA-seq samples, expression profiles can only become more complex. This will cause splits of larger modules. Thus, the full data set with all 24 RNA-seq samples yields the most conservative lower estimate of 14,275 for the number of environmentally responsive genes in large coexpression modules. That means that almost half of the 28,896 annotated genes in *P. pacificus* respond to environmental microbiota.

## Coexpression modules exhibit developmental signatures

Bacterial diets can alter the developmental rate in *C. elegans* and *P. pacificus* [39,47]. Therefore, we wanted to test whether the coexpression modules also show a developmental signature. To this end, we visualized the normalized expression across *P. pacificus* development [43] for all genes in the largest coexpression modules (Fig 3). This showed that the coexpression modules that were obtained from adult worms on different bacteria also exhibit distinct expression profiles during development. Furthermore, this developmental signature is very consistent between most genes of the same module. For example, genes in module 1 are consistently activated only late in development (>48h, Fig 3). Their expression is preceded by genes in module 2 which consistently increase expression starting from the 40h timepoint (Fig 3). This could mean that the same network modules control development as well as the response to environmental microbiota. Alternatively, this developmental signature could be an indirect effect of altered developmental timing. Even if only adult worms were manually picked for RNA extraction, there might still be differences in the age of these worms. This is because worms may delay or accelerate their development in response to different bacteria [39,47] and consequently, adult worms that were grown on different bacteria may not be of the same chronological age. If the expression of certain modules rather follows the chronological age than the morphological stage, such modules could potentially cause a similar developmental signature. However, no matter if the expression profiles represent immediate response to different bacteria or variation in the chronological age of worms, both scenarios would reflect either direct or indirect consequences of the exposure to different microbial environments.

## Coexpression modules have distinct regulatory architecture

The observation of distinct expression profiles among environmentally responsive genes suggests that the coexpression modules might be coregulated by diverse sets of transcription factors. To test this, we searched for overrepresented motifs in the promoter sequences of the largest coexpression modules using a *de novo* motif discovery approach as implemented in the HOMER software [49]. This identified a diverse set of DNA motifs that are highly enriched in specific modules (S6 Fig). While modules 1,5,7, and 23 show enrichments of the same motifs (ZBTB32, CUX1, and LIN54), many other coexpression modules have a unique regulatory architecture with very specific motifs that are only enriched in the given module (e.g. POU5F1 in module 2 and *Foxd3* in module 24, S6 Fig). Unfortunately, it is not straightforward to infer the regulator for a given motif as families of transcription factors might be large and may bind

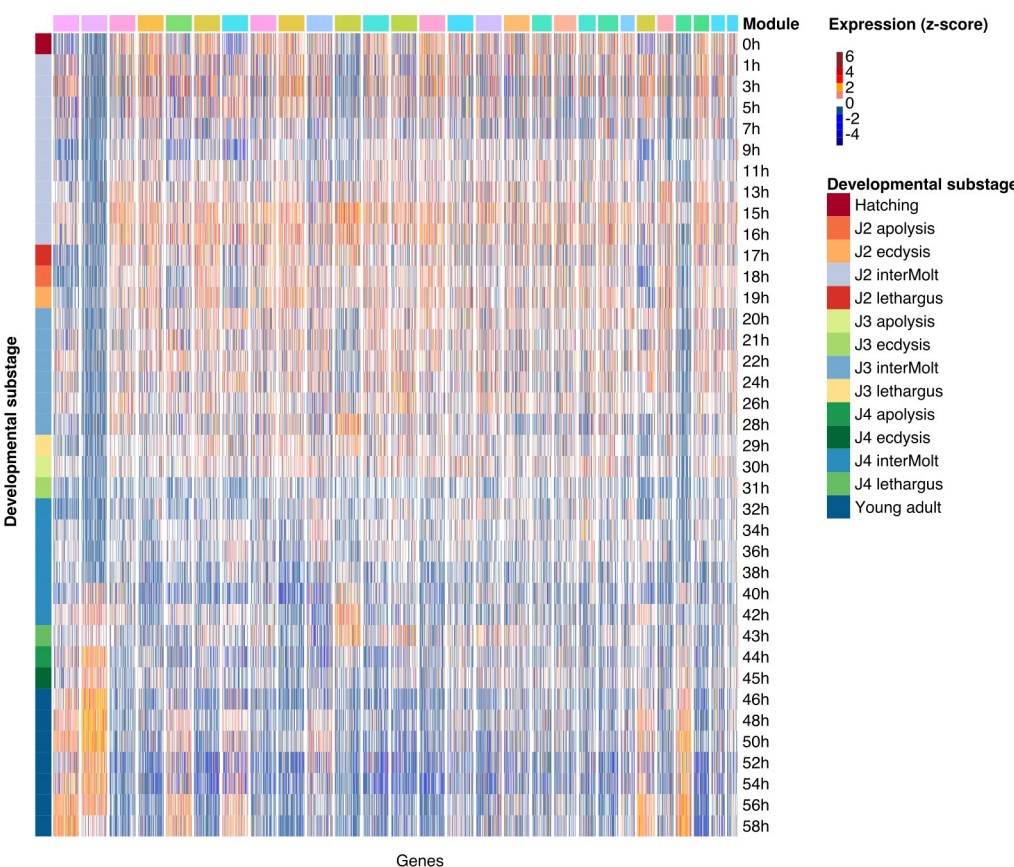

**Fig 3. Developmental signature of coexpression modules.** We visualized the z-score normalized expression levels of genes in the 28 largest coexpression modules throughout postembryonic development after hatching (0h) on *E. coli* OP50 [43]. Modules with more than 100 genes were randomly downsampled to 100 genes. The coexpression modules show distinct expression profiles suggesting a link between environmental response and developmental regulation.

very similar motifs [50]. For example, the human zinc finger and BTB domain containing protein ZBTB32 which binds a motif that is highly enriched in module 1 (S6 Fig) has dozens of orthologs in the genomes of *C. elegans* and *P. pacificus*. However, the most significantly enriched motif of module 2 has the highest similarity with the motif of POU domain homeobox transcription factor POU5F1. This family has only three members (*unc-86*, *ceh-6* and *ceh-18*.) in *C. elegans* [51] and four in *P. pacificus*. Similarly, *C. elegans ces-1* has a one-to-one ortholog in *P. pacificus*. Thus, for individual modules future experimental analysis could be used to dissect the regulatory relationships at a mechanistic level. Nevertheless, already the diversity of motifs that are found across different coexpression modules supports that they have distinct regulatory architecture. In addition, this analysis supports that the genes in a given module are not only coexpressed, but also coregulated.

## 3,727 *Pristionchus pacificus* orphan genes respond to diverse environmental microbiota

To test to what extent new genes could contribute to the response to diverse environmental microbiota, we performed a phylostratigraphic analysis to determine the distribution of gene ages across coexpression modules. Phylostratigraphy is a commonly used method to map gene birth events to a branch within a species tree by searching for the most distant homolog [52].

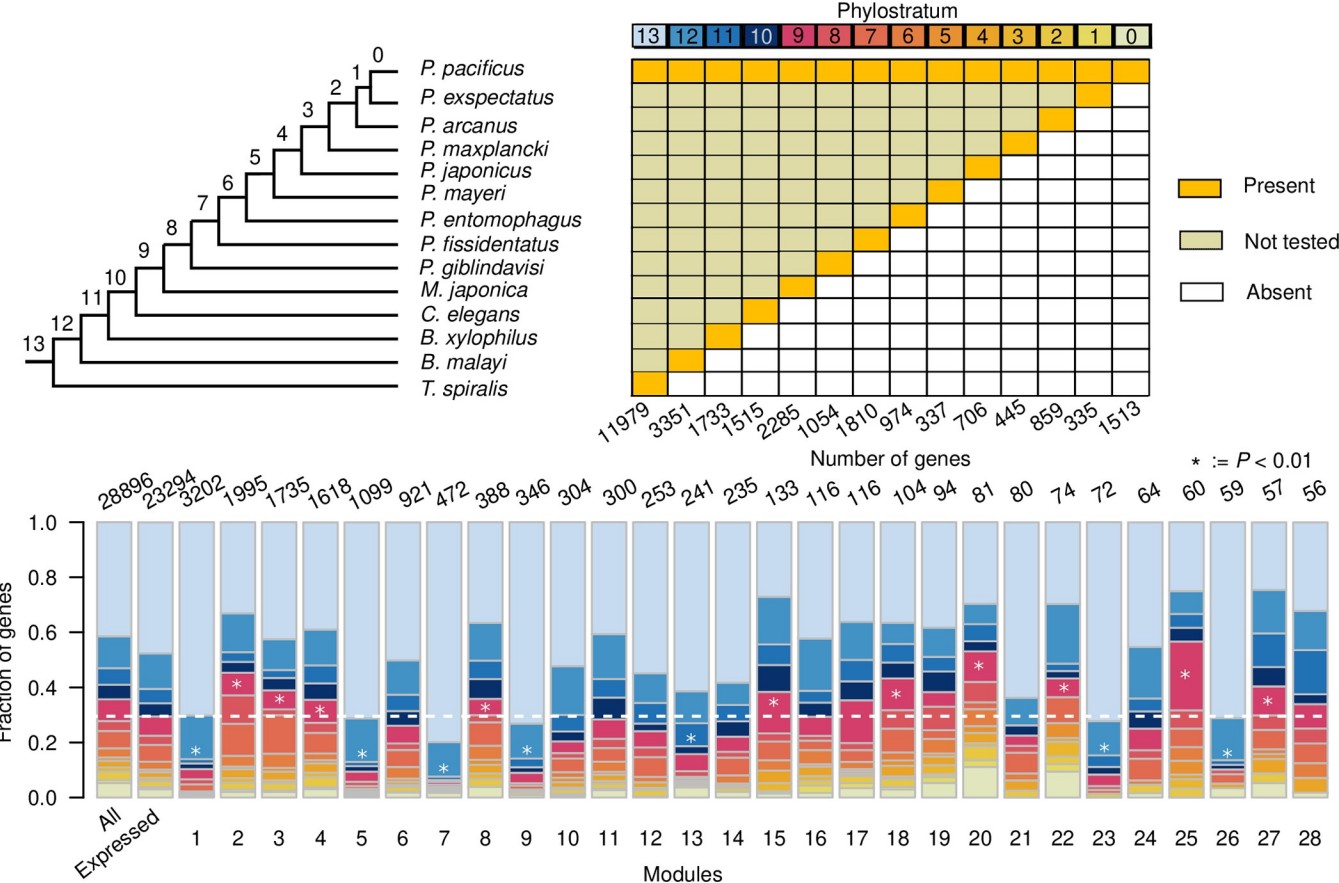

**Fig 4. Phylostratigraphic analysis across coexpression modules.** Phylostrata were defined based on the presence of homologs for a given gene in the most distantly related species. Each phylostratum defines a branch in the phylogeny where a gene likely originated. The barplot shows the distribution of phylostrata across coexpression modules with the dashed line marking the fraction of diplogastrid-specific orphan genes across all genes. The stars indicate significant enrichment and depletion of orphan genes based on simulating the integration of new genes to existing network modules ($P < 0.01$, S7 Fig).

Here, we selected nine additional diplogastrid genomes [30] and the genomes of *C. elegans*, *Bursaphelenchelus xylophilus* [53], *Brugia malayi* [54], and *Trichinella spiralis* [55] to assign genes to phylostrata (S2 Table). Altogether, 10,318 (35.7%) of all genes were defined as diplogastrid family-specific orphan genes (BLASTP e-value $< 10^{-3}$). Visualization of the distribution of phylostrata across the coexpression modules shows that almost half of all the genes in module 2 are diplogastrid-specific orphan genes (Fig 4). Furthermore, among the smaller modules 20 and 22, we observe relatively high ratios of species-specific orphan genes. This finding suggests that the integration of new genes into regulatory networks can happen very rapidly. Alternatively, such genes might represent ancient but rapidly evolving sequences such as antimicrobial peptides where we underestimate the gene age due to the failure to detect homologs [3]. In total, 3,727 diplogastrid-specific orphan genes are found among the 28 largest coexpression modules. As indicated above, this represents a lower estimate for the total number of environmentally-responsive orphan genes as additional orphan genes are found in smaller modules (S4 Fig). Relative to the genome-wide fraction of orphan genes (35.7%), this represents an underrepresentation as only 26.1% of genes in large coexpression modules are orphan genes ($P < 2.2 \times 10^{-16}$, Fisher's exact test). This may be largely explained by the fact that orphan genes are generally lowly expressed [31]. In our data set, only 66.4% of orphan genes are expressed whereas this number increases to 88.5% for non-orphan genes.

Nevertheless, despite their lower level of expression, our study could demonstrate that thousands of orphan genes are indeed embedded into developmental networks that plastically respond to diverse microbiota. Further, it identified which of the orphan genes respond to different environments and it established associations between these orphan genes and specific network modules where they have been embedded.

## The major fast-evolving module is associated with spermatogenesis

The distribution of phylostrata across coexpression modules suggests that new genes are not attached randomly into regulatory networks, i.e. not every module is equally likely to acquire a new gene. On the contrary, individual coexpression modules are more likely to integrate new genes than others. To assess the non-randomness in the distribution of diplogastrid-specific orphan genes, we simulated the integration of new genes into existing networks. Specifically, we tested a model where all network modules are equally likely to acquire a new gene and a model where the probability of attachment is proportional to the size of the module (S7 Fig). Basically, the second model predicts much more accurately the number of orphan genes per module and it allows to define fast evolving modules with significant enrichments of orphan genes (Modules 2–4, 8, 15, 18, 20, 22, 25, and 27, $P < 0.01$, S7 Fig and Fig 4). Constrained modules (Modules 1, 5, 7, 9, 13, 23, and 26) were defined analogously as modules that are significantly depleted in orphan genes. To better characterize the biological differences between such fast evolving and more constrained modules, we performed overrepresentation analysis of protein domains (S3 Table), metabolic pathways (S4 Table), and other expression data sets [43,56–58]. The expression data includes 11 sets of regionally expressed genes that were identified from RNA tomography along the anterior-posterior axis of individual worms [58]. These analyses were complemented by tissue enrichment analysis (TEA, S5 Table) using one-to-one orthologs in *C. elegans* [59]. The overrepresentation of germline associated regions P6-P8 and TEA support that the constrained module 1 is associated with oogenesis (Fig 5A and S5 Table). In contrast, the enrichment of Motile sperm proteins, TEA and expression in sperm-related regions (P5, P9) associate the fast evolving module 2 with spermatogenesis (Figs S8 and 5A and S5 Table). Thus, the two largest environmentally responsive coexpression modules represent a constrained module associated with oogenesis and a fast evolving module that is associated with spermatogenesis. In *C. elegans*, the reproductive system is known to plastically respond to changes of environmental microbiota [60,61]. Thus, it is unsurprising to see the corresponding modules to respond dynamically to diverse bacteria in *P. pacificus*. These results also recapitulate previous findings of signatures of rapid evolution in spermatogenesis-associated genes in *C. elegans* and *P. pacificus* [58,62] This out-of-testis trend of new gene formations was also observed in insects and mammals and likely reflects a combination between rapid evolution of spermatogenesis-associated genes and a more permissive chromatin state [63,64].

## Thousands of orphan genes can be associated with biological processes or tissues

The strong support for the association of the two largest coexpression modules with oogenesis and spermatogenesis motivated us to test if more coexpression modules could be annotated with biologically meaningful labels. Therefore, we complemented protein domain information (S8 Fig), spatial transcriptomics (Fig 5A), and TEA (S5 Table) with the overrepresentation of Kyoto Encyclopedia of Genes and Genomes (KEGG) pathways (Fig 5B) and other *P. pacificus* expression gene sets (Fig 5C) [43,47,56,57,65–67]. This allowed us to assign labels for 22 of the 28 largest coexpression modules (Table 1). Please note that these labels are not exclusive but rather describe the most strongly enriched biological terms. Notably, nine of the coexpression

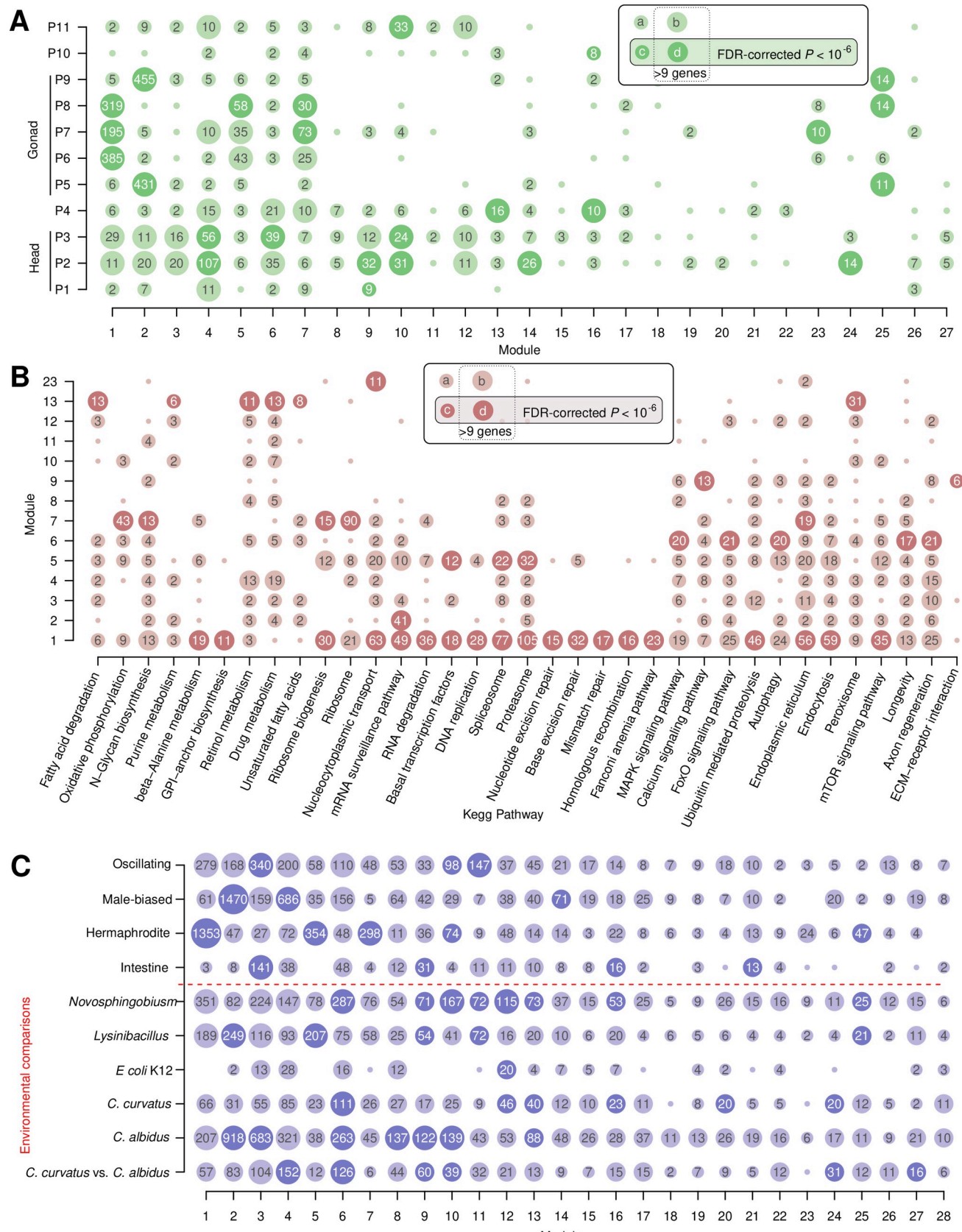

**Fig 5. Transcriptomic and metabolic enrichment of coexpression modules.** (A) Coexpression modules show distinct overlaps with regional genes (P1-P11) that were identified from spatial transcriptomics. (B) Multiple metabolic pathways are enriched in individual coexpression modules. (C) Coexpression modules were compared with multiple expression gene sets. 16 out of 28 coexpression modules show significant overlap with differentially expressed genes in previous comparisons of environmental microbiota.

modules are associated with the nervous system which is also supported by the patterns of G protein coupled receptors (GPCRs), neuropeptides, and nuclear hormone receptors (NHRs) (S9 Fig). All these gene classes are thought to be enriched in neurons [68,69]. In addition, we identified a gland cell related module which captures 17 out of 24 target genes of the regulators of the mouth-form polyphenism in *P. pacificus* [70] (S10 Fig). Altogether, this analysis predicts associations for 13,923 (48%) *P. pacificus* genes with biological processes and anatomical structures. This includes predicted associations for 3,556 (35%) of diplogastrid-specific orphan genes.

## Comparisons with previous RNA-seq studies reveal interactions between environmental microbiota and the nervous system

In the current study, we wanted to get a very broad overview of expression changes across a wide range of bacterial environments. Therefore, we decided to sequence only a single

**Table 1. Biological annotation for coexpression modules.** Most coexpression modules could be labeled with biological processes or tissues based on manual inspection of the results of different overrepresentation analyses.

| Module | Number of genes | Name | Evidence |
|---|---|---|---|
| 1 | 3,202 | Oogenesis 1 | P6-P8, sex-bias, TEA |
| 2 | 1,995 | Spermatogenesis | P5+P9, motile sperm proteins, sex-bias, TEA |
| 3 | 1,735 | Intestine 1 | Intestine RNA-seq |
| 4 | 1,618 | Nervous system 1 | GPCRs, neuropeptides, NHRs, P1-P3, TEA |
| 5 | 1,099 | Oogenesis 2 | P6-P8, sex-bias, TEA |
| 6 | 921 | Nervous system 2 | GPCRs, Axon regeneration, P2-P3, TEA |
| 7 | 472 | Oogenesis 3 | P6-P8, sex-bias |
| 8 | 388 | Nervous system 3 | GPCRs |
| 9 | 346 | Muscle / intestine 2 | Intestine RNA-seq, TEA |
| 10 | 304 | Cuticle 1 | Collagens, oscillation |
| 11 | 300 | Cuticle 2 | Collagens, oscillation, TEA |
| 12 | 253 | Nervous system 4 | GPCRs, NHRs |
| 13 | 241 | Intestine 3 | Fatty acid degradation, drug metabolism, P4 |
| 14 | 235 | Nervous system 5 | GPCRs, P2, TEA |
| 15 | 133 | Unknown | |
| 16 | 116 | Intestine 4 | Intestine RNA-seq, P4 |
| 17 | 116 | Unknown | |
| 18 | 104 | Nervous system 6 | TEA |
| 19 | 94 | Nervous system 7 | TEA |
| 20 | 81 | Orphan 1 | |
| 21 | 80 | Intestine 4 | Intestine RNA-seq |
| 22 | 74 | Orphan 2 | |
| 23 | 72 | Oogenesis4 | P6-P8, TEA |
| 24 | 64 | Gland cell / mouthform | P2, Astacins |
| 25 | 60 | Orphan 3 | |
| 26 | 59 | unknown | |
| 27 | 57 | Nervous system 8 | TEA |
| 28 | 56 | Nervous system 9 | GPCRs |

transcriptome from many different bacterial environments. This is in contrast to previous studies that focussed on specific interactions between hosts and microbes and defined more robust transcriptomic changes with regard to the standard diet *E. coli* OP50 by sequencing multiple biological replicates [47,65–67]. In order to gain additional support for the environmental regulation of the identified coexpression modules, we quantified how many of the coexpression modules exhibit associations with sets of significantly differentially expressed genes in six transcriptomic comparisons of distinct microbial environments including *Novsphingobium*, *Lysinobacillus*, *E. coli* K12, and two *Cryptococcus* yeast strains [47,65–67]. This showed significant enrichments for 16 modules with differentially expressed genes in at least one of the previous studies (Fig 5C). Moreover, we find the intestinal module 9 and the nervous system related module 6 to be most frequently overrepresented. This further supports the impact of different bacteria on the nematode nervous system with potential consequences on behavior. Such an effect has been experimentally characterized in *P. pacificus* for the case of *Novosphingobium* bacteria. Feeding on these bacteria accelerates development and makes the worms more efficient predators of *C. elegans* [47]. Another example of behavioral modulation constitutes the effect of a neurotransmitter release by bacteria on *C. elegans* perception [45]. Notably, six of the nine nervous system related modules are significantly affected in at least one environmental comparison. This highlights the potential of microbial environments to modulate the nervous system and behavior.

## Discussion

Which genes respond to changing environments? Do orphan genes play a role in adapting to diverse microbiota? Can we infer potential functions for orphan genes based on coexpression with known genes? How are new genes integrated into existing biological networks? To study these questions, we characterized the transcriptomic response of *P. pacificus* nematodes to 24 environmental microbiota. This analysis let to the identification of 28 large coexpression modules that contain almost half of all genes in *P. pacificus*. Further integrative analysis associated 22 of these modules with biological processes or tissues. These modules capture previously characterized gene sets such as collagens with oscillating expression [43] or target genes of mouth-form regulators that are expressed in the gland cell [70]. Moreover, these functionally annotated modules contain 3,556 (35%) of diplogastrid-specific orphan genes, of which a large fraction is found in the spermatogenesis associated module 2. While sperm associated genes are known to evolve rapidly [62,71], our study adds evidence that the expression of spermatogenesis-associated genes can be modulated by different microbiota. One major limitation of the current study consists in the fact that we only sequenced a single biological replicate per environment. This strongly limits the usefulness of our data set to dissect the effect of individual bacterial strains on nematode gene expression. However, we would argue that this does not undermine our main findings as coexpression signals can still be inferred from such a data set even if single expression estimates are not robust. In addition, we would like to point out that since the worms have been grown on selected bacteria, the effect of the bacteria likely accumulates in the nematodes throughout development and might be different from an immediate transcriptomic response after short time exposure. Thus, the observed differences could partially represent an indirect consequence of altered developmental rates. Even if only young adults were picked for RNA-seq experiments, we cannot exclude the possibility that the expression of certain gene modules reflects a chronological age rather than a morphological age. Future work could elucidate to what extent the transcriptomic responses represent immediate response to different bacteria or developmental variation. In addition, we currently do not know to what extent different bacteria constitute a diet or could also interact with

nematodes by colonizing their gut [72]. Nevertheless, despite all these uncertainties, we would still argue that most of the observed transcriptomic variation represents either direct or indirect responses to different microbial environments.

One of the main achievements of our work constitutes the association of thousands of *P. pacificus* genes with biological processes and tissues. This includes thousands of diplogastrid-specific orphan genes. Of course, this type of functional annotation represents a completely different level than the knowledge for the two experimentally characterized orphan genes, where knockout and transgenic lines are available that show organism level phenotypes [34,73]. Our functional associations rely on the assumption that coexpression implies cofunctionality. Although this is a frequently employed method for functional assignment [35,36], the example of the out of testis pattern shows that this assumption might not always be true. Specifically, it has been shown that testis-biased expression of many new genes might be an effect of an overall permissive chromatin state leading to higher transcriptional complexity [64]. Thus, our data should rather be considered as a source to generate new hypotheses that have to be tested experimentally. As such, our functional annotations may be helpful to interpret future transcriptomic studies in *P. pacificus*. Some of these modules might also be relevant for studying the polyphenism of the mouth morphology in *P. pacificus*. This phenomenon has developed into one of the best studied animal systems for developmental plasticity [74,75]. Our observation of environmentally dependent expression variation in a gland cell related module that includes many target genes of mouth-form regulators [70] suggests an additional control layer that is independent of the mouth-form. This is because no bacteria are currently known that alter the mouth-form ratio in the highly predatory *P. pacificus* reference strain PS312 [47]. However, it could well be that the same bacteria can induce the predatory morph in a different wild isolate with a low or intermediate frequency of the predatory morph.

Another major finding concerns the integration of new genes into existing gene regulatory networks. Our analysis clearly shows that new genes are not randomly attached to existing modules, but rather specific modules have much higher propensity to acquire new genes. The likelihood to integrate new genes may be associated with the biological function of a given module. For example, while the oogenesis associated modules 1, 5, 7, and 23 are composed of the oldest gene sets, the spermatogenesis-associated module 2 has one of the youngest gene contents. The strong difference in evolvability between these two reproductive processes could be due to strong sperm competition which results from the difference in the number of gametes between both sexes [76]. Another factor could consist in the differential control of transposons between spermatocytes and oocytes resulting in higher potential for molecular innovation during spermatogenesis [77,78]. Another aspect of network evolution concerns the timing of events. The presence of many species-specific orphan genes in some coexpression modules suggests that integration can happen relatively fast. Together with a recent study on the evolution of the polyphenism network [79], our work reveals first insights into the evolution of environmentally responsive networks in *Pristionchus* nematodes and similar studies in other taxonomic groups could be done to test whether these observations reflect general patterns of gene-regulatory network evolution.

## Methods

### Bacterial Culture Conditions

All bacterial strains were recovered from glycerol stocks [42] and then plated on nematode growth medium (NGM) and incubated overnight at 37˚C. From these plates single colonies were seeded in lysogeny broth (LB) medium and grown overnight at 37˚C in a shaking incubator.

## Nematode Culture Conditions

The wild type strain of *P. pacificus* (PS312) was maintained at 20˚C on nematode growth medium (NGM) seeded with *E. coli* OP50 before use in experiments [80]. From every generation, five young adults were transferred to fresh plates with a wormpick.

## RNA sequencing

Bleaching was used to synchronize *P. pacificus* nematodes and to remove *E. coli* OP50 before transferring bleached eggs to NGM plates with bacterial strains [80]. These bacterial plates were obtained by spotting 50 μL of overnight bacterial cultures onto each 6 cm NGM plate with an L-spreader followed by incubation for 2 days. The concentration of the overnight bacterial cultures was quantified by measuring the optical density at 600 nm (OD600). To make the OD600 of the initial cultures identical, the OD600 of the overnight cultures were measured and diluted to OD600 = 1 with fresh LB. Worms were grown on these plates at 20˚C and every generation young adult worms were passaged to fresh bacterial plates. This was done for at least two generations. From mixed-stage cultures, 50 young adults with only one egg were manually selected with a wormpick under a binocular microscope into an Eppendorf tube containing 100 μl M9 buffer and immediately frozen at -80˚C. Total RNA was extracted using Direct-Zol RNA Mini prep kit from Zymo Research according to the manufacturer's guidelines. RNA libraries were prepared using the Illumina Truseq RNA library prep kit according to the manufacturer's guidelines. The libraries were quantified using a combination of Qubit and Bioanalyzer (Agilent Technologies) and normalized to 2.5 nM. Samples were sequenced as 150 bp single end reads on multiplexed lanes of an Illumina HiSeq3000 in our inhouse sequencing facility. Raw reads were depositted at the European Nucleotide archive under the study accession PRJEB60166.

## Developmental timing

Bacterial strains were grown overnight from a single colony in LB medium. The medium was shaken at 180 rpm and incubated at 30˚C for LRB89 and LRB104 and 37˚C for OP50. These cultures were spotted on 6-cm NGM plates and were grown for 2 days at RT. Synchronized J2 worms were obtained by bleaching and were put on these NGM plates [80]. Nematode cultures were grown at 20˚C. 56h later, the distribution of developmental stages was scored based on the vulval development under a ZEISS SteREO Discovery microscope [47].

## Identification of coexpression modules

The raw reads were aligned to the reference *P. pacificus* genome (version El Paco) with STAR (version 2.7.3a) and quantified with featureCounts from the Subread R package (version 2.0.1) using the current gene annotations (El Paco gene annotations version 3) [81–83]. After prefiltering the count matrix by removing genes (rows) that have less than 10 reads total, we retained 23,294 (80.6%) of all *P. pacificus* genes with some evidence of expression. The read counts across the different conditions were normalized with the DESeq2 counts function (with normalized = TRUE option) [84]. The normalized read counts were used to create a coexpression network with MCL [48]. Different correlation and inflation thresholds were assessed for the coexpression network, ranging from 0.6 to 0.8 and from 1.5 to 4 respectively. The network was constructed using a correlation of 0.7 and an inflation threshold of 2. The network modules containing more than 50 genes were selected for further analysis. To assess the robustness of module assignments, we recomputed coexpression networks after either removing the outlier LRB104 or systematically downsampling the data set. Taking the full coexpression network

calculated from 24 RNA-seq samples as reference, we evaluated the classification of 10,000 randomly chosen gene pairs from a subsampled RNA-seq data set. For the full and the subsampled data set, a gene pair was either classified to be part of the same module or not. If a gene pair was classified as being part of the same module in both data sets, such a pair was classified as true positive. If a pair was only part of a module in the subsampled data set, but not in the full data set, this was scored as false positive. True negative and false negative cases were defined analogously. Subsequently, the positive predictive value (PPV) and negative predictive value (NPV) for 10 randomly subsampled data sets of a fixed size (S5 Fig). For comparison, the data set without LRB104 yielded a PPV of 84% and an NPV of 98%.

## Motif analysis and phylostratigraphy

To test for overrepresented motifs in the promoter regions, we focused on the 500-bp nucleotide sequence upstream of each annotated gene in a given module. These sequences were taken as input for the findMotif.pl script of the Hypergeometric Optimization of Motif EnRichment (HOMER) suite (version v4.10.1) [49]. As a background set, we used the 500-bp upstream sequences of all the other large coexpression modules. For visualization, we selected among the significant motifs only one representative motif per known motif match and applied a cutoff that a motif needs to be present in at least 20% of the promoter regions of a given module. For analyzing the distribution of phylostrata across coexpression modules, we assigned a gene to a phylostratum based on the presence of the most distant homolog. For this purpose, we performed BLASTP searches (e-value < 0.001) of all 28,896 *P. pacificus* proteins against protein data of nine diplogastrid genomes (version PPCAC) [30], *C. elegans*, *B. xylophilus*, *B. malayi*, and *T. spiralis* (WormBase ParaSite version WBPS14) [85]. This identified 10,318 diplogastrid specific orphan genes. To simulate the integration of new genes into existing network modules, we first defined the ancestral module sizes based on the number of ancient genes (non-orphan genes) for the 28 large modules. We then simulated the integration of new genes by either assigning equal probabilities for attachment to all network modules or by scaling the attachment probability as a linear function of the module size. The second model predicts much more accurately the number of orphan genes per module (Pearson's r = 0.62, S7 Fig). Therefore, we used 100 iterations of the second model to calculate empirical P-values for the overrepresentation of orphan genes ($P < 0.01$, Figs S7 and 4).

## Compilation of *P. pacificus* expression gene sets

Sets of 3,502 *P. pacificus* genes with regional expression were extracted from supplementary table S3 in [58]. These genes exhibit enriched expression in at least one out of eleven regions (P1-P11) that were identified by RNA tomography of adult *P. pacificus* animals. A conversion table between assembled transcripts (version trinity 2016) and current gene annotations (El Paco gene annotations V3) were obtained from the supplementary table S10 in [43]. We extracted the set of 2,964 developmentally oscillating genes from the supplementary table S3 in [43]. In addition, we reanalyzed *P. pacificus* RNA-seq data sets to compile candidate genes with intestinal expression [56], sex-biased expression [57], and differentially expressed genes in response to altered microbial environments [47,65–67]. For the intestinal data and exposure to *Lysinibacillus*, we identified candidate genes by aligning RNA-seq reads to the *P. pacificus* reference assembly (version El Paco) with the help of the tophat alignment program (version v2.0.14, default settings) and testing for differential expression using the cuffdiff program (version v2.2.1, $P<0.1$) [86]. This yielded 723 intestine enriched genes, and 1,959 candidate genes with differential expression after one hour exposure to *Lysinibacillus*. For the data sets with multiple biological replicates [47,57,65,67], we realigned RNA-seq data with STAR (version

2.7.1a) [81], generated count matrices with the featureCounts function of the subread package in R (version 3.6.3) [82] and for significant differential expression with the DESeq2 package (FDR-corrected $P<0.05$) [84].

## Overrepresentation of gene families, metabolic pathways, and expression sets

To test for overrepresentation of specific gene sets among the coexpression modules, we first generated protein domain annotations by the hmmsearch program (version 3.3, -E 0.001 option) against the Pfam-A.hmm database (version 3.1b2). Similarly, KEGG annotations were obtained by identification of orthologs in the KEGG database using the blastkoala web application (with the 'eukaryotes' and 'family_eukaryotes' values for taxonomic group and database level, respectively) [87]. Genes with orthologs in the KEGG database were then annotated with the corresponding *C. elegans* KEGG accessions. From the protein domain predictions, we identified 1,434 genes with protein domains that were termed as GPCRs and 275 putative NHR genes based on the presence of the Hormone_recep domain (PF00104). Potential neuropeptides were identified by BLASTP search (-evalue 0.001) of 107 *C. elegans* neuropeptides (*flp* and *nlp* genes) against the current *P. pacificus* proteins. For 59 *C. elegans* neuropeptides, we could identify 85 homologs in *P. pacificus* of which we only used 38 single copy candidates as neuropeptide candidates. These annotations were combined with various expression data sets to perform overrepresentation analysis using the Fisher's exact-test with multiple testing correction (FDR corrected $P < 10^6$) in R (version 3.6.3).

## Annotation of coexpression modules

In order to associate the coexpression modules with biological processes or tissues, we complemented the results of the overrepresentation analysis in *P. pacificus* with tissue enrichment analysis (TEA) of *C. elegans* orthologs [59]. Specifically, *C. elegans* orthologs of *P. pacificus* genes were extracted from the best reciprocal BLASTP hit data set from Athanasouli *et al.* (2020) [83] and submitted to the WormBase web interface (https://wormbase.org/tools/enrichment/tea/tea.cgi, version WS283) [83]. We manually inspected the results of all different analyses and subjectively assigned biological processes or tissues for a given module. Functional assignments with multiple types of evidence (e.g. Protein domains, spatial transcriptomics, sex-biased expression) should be considered as high confidence associations whereas associations that are supported by only a single analysis are more uncertain.

## Supporting information

**S1 Fig. Analysis of the most similar and dissimilar RNA-seq data sets.** The scatter plots show the normalized expression (TPM) for different pairs of samples. As lowly expressed genes tend to be more variable, we visualized the number of genes with at least two-fold expression difference across multiple mean expression levels.
(PDF)

**S2 Fig. Parameter combinations for MCL clustering.** We evaluated the total number of modules and singleton modules as a function of the inflation factor (I) and correlation coefficient (r). High r and I values generally increase the number of modules, whereas low r and I parameters generate fewer but larger modules. We decided to use r = 0.7 and I = 2 for the final analysis, because it gave a moderate number of modules with a relatively low number of singletons.
(PDF)

**S3 Fig. Developmental timing on different bacteria.** The distribution of developmental stages 56h after J2 synchronization is shown for *P. pacificus* worms on *E. coli* OP50 and two *Wautersiella* bacteria (5 biological replicates). Nematodes grow slower on both *Wautersiella* strains. The significance level was computed by a $\chi^2$-test (mean *P*-value from all pairwise comparisons).
(PDF)

**S4 Fig. Comparison of MCL networks constructed with or without LRB104.** The heatmap shows fraction genes from modules of the complete coexpression network (including LRB104) that overlap within a given module from the coexpression network without LRB104. For most modules there exists a 1–1 correspondence between both networks, indicating that the network structure is robust with regard to the *Wautersiella* LRB104 data set.
(PDF)

**S5 Fig. Comparison of Coexpression networks on subsampled RNA-seq data.** Taking the full coexpression network calculated from 24 RNA-seq samples as reference, we evaluated the classification of 10,000 randomly chosen gene pairs using subsampled RNA-seq data. For the full and the subsampled data set, a gene pair was either classified to be part of the same module or not. This allowed us to calculate the positive predictive value (PPV, panel A) and negative predictive value (NPV, panel B) for 10 randomly subsampled data sets of a fixed size. While the NPV is always close to 1, the PPV shows drastic differences between the full and subsampled data. This suggests that with fewer RNA-seq samples, additional gene pairs are assigned to the same coexpression module. However, with additional samples, such modules may be split into smaller components. Consistently, the number of genes in large modules (N>50) is much higher for smaller sample sizes (panel C). Panel D shows the number of diplogastrid-specific orphan genes in large modules.
(PDF)

**S6 Fig. Regulatory architecture of coexpression modules.** Significantly overrepresented motifs were identified for each module by the HOMER software. We arbitrarily selected motifs that occurred in at least 20% of promoters of a given module and visualized their distribution across all modules. Note that the complementary regulation by less frequent motifs and other regulatory mechanisms such as microRNAs are not considered here. Sequence logos for each motif are shown at the right and the labels to the left indicate the best motif match among known motifs.
(PDF)

**S7 Fig. Enrichment analysis for simulated network evolution.** We simulated the integration of novel genes into existing networks by estimating ancestral module sizes based on the number of ancient genes (non-orphan genes) and then assigning an equivalent number of orphan genes to existing modules with equal probabilities (panel A and C) and with probabilities that were proportional to the module size (panel B and D). The scatterplots show the observed and simulated number of orphan genes per module. The barplots show the median enrichment of the observed relative to the simulated number of orphan genes (error bars indicate the minimal and maximal values from 100 simulations).
(PDF)

**S8 Fig. Overrepresented protein domains in coexpression modules.** Specific protein domains are strongly overrepresented in twelve coexpression modules. The left barplot shows the number of genes with a given protein domain for each module and the the right bar plot shows the negative logarithm of the FDR corrected P-value (Fisher's exact test). The most

significant association is between Motile sperm proteins (PF00635) and coexpression module 2.
(PDF)

**S9 Fig. Distribution of GPCRs, NHRs, and neuropeptides across the coexpression modules.**
(PDF)

**S10 Fig. Expression of selected module 24 genes.** The heatmap shows the expression of genes that are shared between module 24 and the target genes of mouth form regulators and additional module 24 genes with 1–1 orthologs in *C. elegans*. The *C. elegans* ortholog of PPA25527 (D1044.3) is reported to be expressed in the gland cell and reporter lines of multiple candidate genes show expression in the gland of *P. pacificus* (Sieriebriennikov et al. 2020) [70].
(PDF)

**S1 Table. Coexpression modules.**
(XLSX)

**S2 Table. Phylostrata.**
(XLSX)

**S3 Table. Protein domain information.**
(XLSX)

**S4 Table. *Pristionchus pacificus* KEGG annotation.**
(XLSX)

**S5 Table. Tissue enrichment analysis (TEA).**
(XLSX)

## Acknowledgments

We would like to thank all members of the Sommer lab for helpful discussions.

## Author Contributions

**Conceptualization:** Christian Rödelsperger.

**Data curation:** Marina Athanasouli, Christian Rödelsperger.

**Formal analysis:** Marina Athanasouli, Christian Rödelsperger.

**Investigation:** Nermin Akduman, Waltraud Röseler, Penghieng Theam.

**Methodology:** Marina Athanasouli, Christian Rödelsperger.

**Project administration:** Marina Athanasouli, Christian Rödelsperger.

**Supervision:** Christian Rödelsperger.

**Visualization:** Marina Athanasouli, Christian Rödelsperger.

**Writing – original draft:** Marina Athanasouli, Christian Rödelsperger.

**Writing – review & editing:** Christian Rödelsperger.

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
