## [Decision Letter · Decision Letter 0]

14 Apr 2023

Dear Dr %Rödelsperger%,

Thank you very much for submitting your Research Article entitled 'Thousands of Pristionchus pacificus orphan genes were integrated into developmental networks that respond to diverse environmental microbiota' to PLOS Genetics.

The manuscript was fully evaluated at the editorial level and by independent peer reviewers. The reviewers appreciated the attention to an important problem, but raised some substantial concerns about the current manuscript. Based on the reviews, we will not be able to accept this version of the manuscript, but we would be willing to review a much-revised version. We cannot, of course, promise publication at that time.

If you decide to revise the manuscript for further consideration at PLOS Genetics, please aim to resubmit within the next 60 days, unless it will take extra time to address the concerns of the reviewers, in which case we would appreciate an expected resubmission date by email to plosgenetics@plos.org.

We are sorry that we cannot be more positive about your manuscript at this stage. Please do not hesitate to contact us if you have any concerns or questions.

Yours sincerely,

Kaveh Ashrafi

Academic Editor

PLOS Genetics

Gregory Barsh

Editor-in-Chief

PLOS Genetics

Reviewer's Responses to Questions

**Comments to the Authors:**

Reviewer #1: This article may represent a significant step toward understanding orphan gene function in P. pacificus and possibly beyond (including the nematode model C. elegans), which should accelerate their experimental study and elucidation. The authors provide a creative roadmap for exploring orphan gene function in other organisms, by exploiting now widely available transcriptomics technologies and datasets. They also provide a unique open resource that can be mined by other groups for their own investigations. Lastly, the authors provide essential insight into the sexually dimorphic evolution of gene regulatory networks, which resonates well beyond nematode phyla. The article is very well written and well-presented, making it an easy and interesting read.

The broader idea behind the authors approach is reminiscent of the C. elegans gene expression mountain database that was used successfully in earlier years to narrow-down investigation of gene functions and genetic interactions by C. elegans biologists. Thanks to the authors, other researchers will be able to focus the experimental investigation of diplogastrid orphan genes around specified predicted functions. This may alleviate a key bottleneck in attributing function to orphan genes, and thus improve the depth of annotation for less conserved genetic sequences, especially in newly sequenced divergent genomes.

Major limitation.

While the authors analysis is innovative and convincing, the dataset itself is limiting as the authors do not appear to have performed biological replicates for the conditions studied, instead relying on sequencing 24 transcriptome of young P. pacificus adults grown for two generations on 24 different monoxenic bacterial cultures. While the methodologies indicate that 50 adults per sample were individually picked checking that they were visually (or morphologically as stated by the authors) at the same developmental stage (holding a single egg), the apparent absence of biological replicates in this analysis (by contrast to collecting triplicates or tetraplicates in other studies) undermines the strength of the study’s conclusions. Given the observed variability in both worm and bacterial growth across biological replicates in other studies and their possible impact on transcriptome expression, the lack of biological replicates here means that we could be looking here at effects that are specific to this dataset and cannot be easily reproduced.

While this undermines the usefulness of this dataset in understanding how given bacterial diets may specifically influence P. pacificus gene expression, it would not majorly affect the prediction of biological functions associated with orphan gene clusters, nor would it diminish the interest of the analytical approach chosen.

Although the authors initially (in their results) propose interpretations that, one may argue, go beyond what can be reasonably concluded from their results, the discussion is very sensible and highlights key limitations of the study, thus providing a fair and insightful account of their research.

They further highlight the fact that the main value of the study lies as a resource and a predictive tool to guide future investigations of orphan genes, which I agree with. Beyond that, the various ways they analyzed their data and compared them to other published datasets has clear merit and will be a source of inspiration for others.

Specific issues

1) Some methods are minimal, particularly for sample preparation, and would not allow for replication of the findings. Additional references to the standard protocols used, and details about the sequencing facility used (one cannot assume that an Illumina platform is standard equipment, yet) are necessary.

2) The authors should also be careful in their use of the word nematode (generalization) in lieu of P. pacificus. This is particularly problematic when looking at interactions with bacteria as adult P. pacificus are parasites and not bacterivorous predators, by contrast with C. elegans for instance. C. elegans adult responses to bacteria likely reflect influences from both a dietary and a gut microbiota component. Generalizing statements to “nematodes” that include species with distinct ecologies is inaccurate.

3) Lines 298-299: « suggesting that the transcriptomic response of nematodes to various environmental microbiota do not strictly reflect phylogeny ». This is seemingly true for P. pacificus but perhaps misleading as it could be that strain-specific changes are obscuring phylum specific changes (a greater coverage of bacterial diversity might reveal that). This could also be due to a methodological limitation: the limited ability to detect changes led to overlooking phylum-specific changes.

Most of the variation seem to come from a couple of isolates LRB104, 80 and 17, which may dwarf other changes and lower the discovery rate of other interesting effects. Would the authors consider rerunning the analysis excluding at least LRB104?

Fig2, could the authors also provide a reordered ranking of bacterial list based on phylogeny to help visualizing whether one or more modules may change expression based on bacterial phylogeny? Not all transcriptional changes may be equal in terms of biological relevance and key modules that could be related to known immune or metabolic pathways might actually map phylogeny when others don’t. The whole transcriptome picture may be masking that.

4) Could the authors specify in the fig 3 legend and/or the text what control condition they used to monitor gene expression in the 28 modules over time? I assume it is E. coli OP50 but it needs to be made clear there for non-nematologists. The authors should also specify the nematode species in the legend. It would also be very interesting to identify the timing of larval stages either on the left or the right of the Fig3 plot.

5) Fig4, how about reordering modules based on similarity in regulatory architecture? 1, 5, 7 and 23 for instance will clearly appear correlated. While I appreciate the unbiased approach, have the authors considered looking for known and functionally characterized regulatory motifs associated with major transcription factors to see how they cluster across their modules? Note that some motifs might be engaged successively within a module because the module includes transcription factors and microRNAs acting within the same pathway, and these may be missed with the 20% threshold applied. Perhaps a comment on that cold be helpful.

6) Fig 5. Interesting representation. About the expression “gene age class”, pardon my candor here but is it so commonly used that it cannot be said differently? My issue with the phrase is that it resemble “age genes”, which characterizes genes modulating aging. Could it be referred to as “gene ancestry class” instead?

7) The authors state line 349 that « the integration of new genes into regulatory network can happen very rapidly ». This would not be particularly surprising for once, but secondly, the evolutionary distance considered here is pretty long (likely representing over 200 million years) as nematodes are an ancient and extent phylum. If this were to be considered from a mammalian evolution perspective, it would be a slow process. Rather than stating “very rapidly”, “relatively rapidly” may be preferred.

8) Figure 7C, “maile-biased” instead of “male-biased”

9) Sperm protein encoding genes tend to regularly pop up as very significantly modulated in C. elegans transcriptomic studies. The significance of these changes is mostly understudied, and apart from the authors’ previous work, has also been attributed to differences in biological age between the conditions tested that careful sampling did not manage to avoid. This is particularly believable when comparing worms grown on distinct diets and/or bacteria as these often affect reproductive timing more than other physiological functions, leading to a different developmental timing for reproductive organs vs other major organs. The fact that these genes appear as one of the major orphan gene clusters revealed in this study, is thus unsurprising. The authors do acknowledge this in conclusion but perhaps it could also come across in their result section.

Reviewer #2: The study described in the manuscript takes an interesting approach to assigning functional annotations, even if putative, to orphan genes unique to the parasitic model nematode Pristionchus pacificus. The premise of using expression under exposure to different bacteria as a proxy for function makes sense. The described analysis enables exploration of ideas about the creation of new genes and function and their incorporation into regulatory networks and offers a resource that can be utilized as a starting point for studying the function of otherwise uncharacterized genes. With that said, some data is over-interpretated, assigning significance (without statistical support) to patterns that might arise randomly, and also, in some cases, too much information, including figures that are more suitable for supplementary information (i.e. Figs. 6 and 4). Detailed comments are described below:

1) It’s not clear to me why the authors would start with a paragraph that global transcription profiles of P. pacificus worms raised on different bacteria are essentially similar. Considering that subsequent analyses focus on differences between gene responses to different bacteria, this runs the risk of confusing the reader. Differences of interest, which are described later on, are not expected to be discerned at the global scale. In fact, in the second section the authors write “Thus, almost half of the 28,896 annotated genes in P. pacificus respond to environmental microbiota.” If so, how come the global correlations were so high (0.9)?

2) The identification of modules of co-expressed genes is a central element in this study. The authors demonstrate that the distinction between these modules is robust so that removal of the data for the most strongly affecting bacteria LRB104 does not change module membership.

Repeating this with other expression profiles – removing data for different strains and recomputing the graph - could lend further confidence in the described model.

3) In fig. 3 the authors suggest that developmental signatures in co-expression modules on different bacteria may represent effects of different bacteria on developmental timing, but the authors indicated earlier that all worms harvested were adults of the same, relatively precise stage. If so, developmental delays are not an option here. This runs the risk of confusing the reader.

4) Fig. 5 is particularly important for the analysis, as it describes that relationships between orphan genes and co-expression modules. In attempting to establish a relationship I see no statistical evaluation. The authors report that 3727 orphan genes are included among the co-expressed genes but considering that a third of P. pacificus genes are considered orphans, and that co-expressed modules included 14,275 genes, 3727 seems like an under-representation of orphan genes among the co-expressed genes, so there is nothing particularly special about their inclusion in the modules. This could happen by chance. So, it’s not clear what’s the point the authors are trying to make here. Is the point that orphan genes are part of regulatory networks in the first place? If so, isn’t that the expectation for de novo created genes, which are thought to arise (at least in some cases) through the evolution of regulatory sequences (Van Oss 2019)? This needs to be clarified.

5) “The distribution of age classes across coexpression modules suggests that new genes are not attached randomly into regulatory networks, i.e. not every module is equally likely to acquire a new gene.” – The authors should provide statistical support for this claim of non-random distribution, or avoid it. The data does not rule out that new genes are in fact attached randomly into regulatory networks.

6) Why is module 1 defined as “highly constrained”? what does that mean? And what is the “fast evolving module”? no number is mentioned.

7) Figures 4 and 6 don’t seem to add useful information, or such that is necessary to better understand the data. They are better off moved to supplementary material.

**Have all data underlying the figures and results presented in the manuscript been provided?**

Reviewer #1: Yes

Reviewer #2: Yes

PLOS authors have the option to publish the peer review history of their article (what does this mean?). If published, this will include your full peer review and any attached files.

Reviewer #1: No

Reviewer #2: No

---

## [Decision Letter · Decision Letter 1]

7 Jun 2023

Dear Dr %Rödelsperger%,

Thank you very much for submitting your Research Article entitled 'Thousands of Pristionchus pacificus orphan genes were integrated into developmental networks that respond to diverse environmental microbiota' to PLOS Genetics.

The manuscript was fully evaluated at the editorial level and by independent peer reviewers. The reviewers appreciated the attention to an important topic but identified some  minor concerns that we ask you address in a revised manuscript.

We therefore ask you to modify the manuscript according to the review recommendations. Your revisions should address the specific points made by each reviewer.

Yours sincerely,

Kaveh Ashrafi

Academic Editor

PLOS Genetics

Gregory Barsh

Editor-in-Chief

PLOS Genetics

Reviewer's Responses to Questions

**Comments to the Authors:**

Reviewer #1: I appreciate the extra explanations and supporting information provided, which I believe has clarified key points and improved the manuscript. The revised version has now addressed all comments raised, although the limitation of dealing with single biological replicates remains.

While the authors have reported and discussed this limitation, it may be worth reiterating it in the abstract to not mislead (careless) readers:

" Specifically, we performed RNA-seq experiments of P. pacificus worms grown on monoxenic cultures of 24 different bacteria" could be rephrased as follows: "Specifically, we analysed 24 RNA-seq samples from adult P. pacificus worms raised on 24 different monoxenic bacterial cultures".

I also have a couple of additional points that I would like to be addressed before publication:

It is customary to provide a general method section that describes the way worms and bacteria are routinely maintained.

There are several instances of abbreviations used in Tables and Figures that are not spelled out in the legends. While some specialist may know exactly what they mean, the broader readership of PLoS Genetics would not.

Finally, there are issues with the way the pdf file comes out, so it would be worth ascertaining that the resolution of the figures is as intended before publishing it.

I trust that all transcriptomics datasets (raw reads) will be made available for others to study.

Reviewer #2: The revisions made in the new version address my earlier concerns. Two small comments:

1 .It would be less confusing in Fig. 3 to distinguish the colors for modules from those for developmental stages.

2. In the abstract, line 50: “higher” than what? Could the author mean just “high”? or, “higher than expected by chance”?

**Have all data underlying the figures and results presented in the manuscript been provided?**

Reviewer #1: Yes

Reviewer #2: Yes

PLOS authors have the option to publish the peer review history of their article (what does this mean?). If published, this will include your full peer review and any attached files.

Reviewer #1: No

Reviewer #2: No

---

## [Editor Report · Decision Letter 2]

15 Jun 2023

Dear Dr %Rödelsperger%,

We are pleased to inform you that your manuscript entitled "Thousands of Pristionchus pacificus orphan genes were integrated into developmental networks that respond to diverse environmental microbiota" has been editorially accepted for publication in PLOS Genetics. Congratulations!

Yours sincerely,

Kaveh Ashrafi

Academic Editor

PLOS Genetics

Gregory Barsh

Editor-in-Chief

PLOS Genetics

Comments from the reviewers (if applicable):

**Data Deposition**

http://datadryad.org/submit?journalID=pgenetics&manu=PGENETICS-D-23-00215R2

**Press Queries**

---

## [Editor Report · Acceptance letter]

26 Jun 2023

PGENETICS-D-23-00215R2 

Thousands of Pristionchus pacificus orphan genes were integrated into developmental networks that respond to diverse environmental microbiota 

Dear Dr Rödelsperger, 

We are pleased to inform you that your manuscript entitled "Thousands of Pristionchus pacificus orphan genes were integrated into developmental networks that respond to diverse environmental microbiota" has been formally accepted for publication in PLOS Genetics! Your manuscript is now with our production department and you will be notified of the publication date in due course.

With kind regards,

Anita Estes

PLOS Genetics

On behalf of:
